# Understanding the role of physical activity on the pathway from intra-articular knee injury to post-traumatic osteoarthritis disease in young people: a scoping review protocol

Karl Morgan [1,2,3,4] James Cowburn,[1,2] Matthew Farrow,[1,3] Josh Carter,[1,2] Dario Cazzola,[1,2,5] Jean-Philippe Walhin,[1,3] Carly McKay[1,2,4]

¹Department for Health, University of Bath, Bath, UK
²University of Bath, Centre for Health and Injury and Illness Prevention in Sport (CHI2PS), Bath, UK
³University of Bath, Centre for Nutrition and Exercise Metabolism (CNEM), Bath, UK
⁴Centre for Sport, Exercise and Osteoarthritis Versus Arthritis, University of Bath, Bath, UK
⁵University of Bath, Centre for the Analysis of Motion, Entertainment Research and Applications (CAMERA), Bath, UK

**Correspondence to**
Karl Morgan;
kdm40@bath.ac.uk

## ABSTRACT

**Introduction** The prevalence of intra-articular knee injuries and reparative surgeries is increasing in many countries. Alarmingly, there is a risk of developing post-traumatic osteoarthritis (PTOA) after sustaining a serious intra-articular knee injury. Although physical inactivity is suggested as a risk factor contributing to the high prevalence of the condition, there is a paucity of research characterising the association between physical activity and joint health. Consequently, the primary aim of this review will be to identify and present available empirical evidence regarding the association between physical activity and joint degeneration after intra-articular knee injury and summarise the evidence using an adapted Grading of Recommendations Assessment, Development and Evaluations. The secondary aim will be to identify potential mechanistic pathways through which physical activity could influence PTOA pathogenesis. The tertiary aim will be to highlight gaps in current understanding of the association between physical activity and joint degeneration following joint injury.

**Methods** A scoping review will be conducted using the Preferred Reporting Items for Systematic reviews and Meta-Analyses extension for Scoping Reviews checklist and best-practice recommendations. The review will be guided by the following research question: what is the role of physical activity in the trajectory from intra-articular knee injury to PTOA in young men and women? We will identify primary research studies and grey literature by searching the electronic databases Scopus, Embase: Elsevier, PubMed, Web of Science: all databases, and Google Scholar. Reviewing pairs will screen abstracts, full texts and will extract data. Data will be presented descriptively using charts, graphs, plots and tables.

**Ethics and dissemination** This research does not require ethical approval due to the data being published and publicly available. This review will be submitted for publication in a peer-reviewed sports medicine journal irrespective of discoveries and disseminated through scientific conference presentations and social media.

**Trial registration number** https://osf.io/84pnh/.

## STRENGTHS AND LIMITATIONS OF THIS STUDY

⇒ A scoping review is the most appropriate approach to identify and describe current evidence on the complex relationship between physical activity and post-traumatic osteoarthritis due to the multifaceted aetiology underpinning the condition.
⇒ This novel research lays an important foundation for further empirical research by mapping current evidence in the literature, examining potential mechanistic pathways using a multidisciplinary approach and identifying gaps in current knowledge.
⇒ Due to the broad nature of a scoping review, this approach may limit the ability to comprehensively elucidate pathological mechanisms.

## INTRODUCTION

Knee injuries, resulting from both contact and non-contact mechanisms,[1–9] are a common orthopaedic complaint in young, sporting populations worldwide[10] and often involve damage to multiple tissues inside the joint.[11–14] Reconstruction surgery is commonly used to treat the injured tissue,[9 11 13 15 16] with hundreds of thousands of knee injuries and reparative reconstruction procedures performed worldwide every year.[13 17–19] Current data suggests rates of intra-articular injury and tissue repair are rising in several countries,[17–19] with knee injuries predominantly affecting people under 40 years of age,[13 17–19] which is alarming considering injury greatly increases the risk of developing knee osteoarthritis (OA).[20] Knee OA is a highly prevalent disorder[21] which is becoming more common,[21–23] and is composed of both a disease and an illness component.[24] OA disease is associated with illness,[25] and the threshold for disease inducing illness is low.[24] However, the presence of illness does not always indicate structural deterioration and

BMJ

disease does not always induce symptomology,[26 27] with history of knee injury alone enough to induce illness.[27]

Knee OA affecting people with a history of intra-articular injury is defined as post-traumatic osteoarthritis (PTOA).[28–31] There is 10-fold increased odds of experiencing PTOA disease between 3-10 years following a knee injury.[27] Further, the prevalence of the disease 11–17 years following knee trauma (48–52%)[29 32] is concerning when considering average age at diagnosis for idiopathic knee OA is 55 years,[33] which indicates a typical case of PTOA would increase disease burden duration. After disease diagnosis, the joint may deteriorate to end-stage OA, resulting in increased odds for total knee replacement (TKR), which occurs at a noticeably younger age in populations with an injury history than those without.[34] Knee OA is the primary cause for TKR[35] and the number of TKRs in people aged under 55 is increasing.[36] As TKRs typically last 25 years,[37] these younger patients experiencing PTOA may sustain subsequent prosthesis failure within their lifetime.[38]

In terms of PTOA illness, after intra-articular knee injury or anterior cruciate ligament reconstruction there is a high prevalence of knee symptomology (eg, pain, stiffness, swelling, crepitus) and loss of function.[39–43] Knee pain is described as the cardinal symptom of OA,[44] and often drives people to seek medical care.[45–47] When applying the pain-avoidance model,[48] developed from idiopathic knee OA populations, to PTOA, experiencing disease and illness due to PTOA may exacerbate loss of function across the lifespan, thereby accelerating the decline below the threshold of independence (figure 1).[46 49] Consequently, understanding joint deterioration and characterising risk factors for PTOA in knee injury populations, with a view to prevent or slow condition progression, is paramount.

Physical inactivity is postulated as one of several risk factors that may compound joint degeneration following intra-articular knee injury.[50 51] Subjective and objective measurements of physical activity (PA) levels indicate reduced participation following knee injury[52–60] and many individuals do not return to the same level of sports participation or return to sport at all after anterior cruciate ligament reconstruction.[61–63] Therapeutic exercise is a core treatment for knee OA, and is considered beneficial to joint health,[64] while physical inactivity has been touted as a key contributor to joint deterioration in idiopathic OA.[22 50] However, a recent narrative review suggested there is little minimal robust scientific evidence linking PA level and joint health following injury.[51] Furthermore, although PA often involves knee joint loading (eg, walking or running), there is limited research investigating the association between cumulative joint loading (dictated by PA) and joint health after knee joint trauma. Lastly, there are several physiological maladaptations that can occur after a traumatic knee injury, such as; increased adiposity,[41 54 59 65–67] knee extensor and flexor weakness,[68–70] muscle atrophy,[71 72] intramuscular fat accumulation,[73 74] and bone mineral density reduction,[75] which could influence joint health. However, these associations with PTOA are yet to be fully understood. Considering PA is causally associated with systemic inflammation,[76–78] adiposity,[79] muscle size[80 81] and quality,[81–86] and bone mineral density[87 88] in uninjured populations, exploring the associations between these factors in injured populations may help to explain the complex relationship between PA and joint health following intra-articular knee injury.

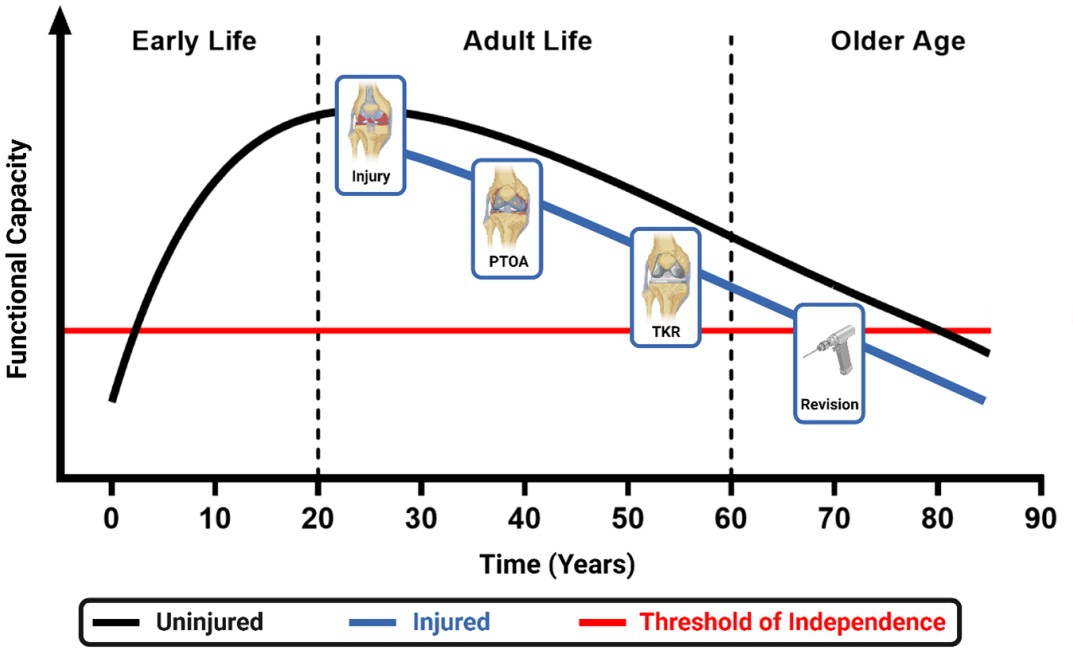

**Figure 1** Hypothetical trajectory of injury to lifelong disease burden. PTOA, post-traumatic osteoarthritis; TKR, total knee replacement.

This protocol documents the procedure for conducting the review. The primary aim of the scoping review will be to identify and present available empirical evidence regarding the association between PA and joint degeneration after intra-articular knee injury and summarise the evidence using an adapted Grading of Recommendations, Assessment, Development and Evaluations (GRADE).[89–91] The secondary aim will be to establish potential mechanistic pathways through which PA could influence PTOA pathogenesis. The tertiary aim will be to highlight gaps in current understanding of the association between PA and joint health following injury.

## METHODS

This scoping review is an exploratory project which will systematically map the literature relevant to the research question, identify key concepts, theories and sources of evidence and gaps in the research.[92 93] This protocol has been prepared to ensure the transparency of the review process and will be used as a guide when conducting the review and reporting the outcomes to minimise the risk of reporting bias.[94 95] Further, this protocol has been developed in accordance with the adapted Preferred Reporting Items for Systematic Review and Meta-Analysis Protocol (PRISMA-P)[96 97] for scoping reviews[95] (online supplemental material 1). This review is registered on Open Science Framework (https://osf.io/84pnh/) and the final output will be underpinned both by the Preferred Reporting Items for Systematic Reviews and Meta-Analyses extension for Scoping Reviews[98] and best practice recommendations.[99–101] Deviations from

this protocol when conducting the review will be documented and published in supplementary material of the final output. This scoping review is currently in progress, with the first search completed in August 2022 and has a projected end date of May 2023.

### Patient and public involvement

During protocol design and planning, there was no patient or public involvement.

### Framework

This review will follow the Arksey and O'Malley framework and adhere to the Joanna Briggs Institute quality assessment recommendations (figure 2).[99 102]

### Stage 1: identifying the research question

The review will be guided by the following research question: What is the role of PA on the trajectory from intra-articular knee injury to PTOA in young men and women? As PTOA is a joint condition with a 'disease' component and an 'illness' component,[24] we will only describe the association between PA, and the factors PA underpins, with PTOA disease in this review. Disease will be defined as: 'abnormalities of the structure and function of body organs and systems that can be specifically identified and described by reference to certain biological, chemical or other evidence'.[103] Additionally, we will only include studies working with young participants (average age of study population at study start or follow-up measurement is 18–40 years old) in this review. Within the research question, the following themes have been identified *a priori*: PA, systemic inflammation, knee joint load,

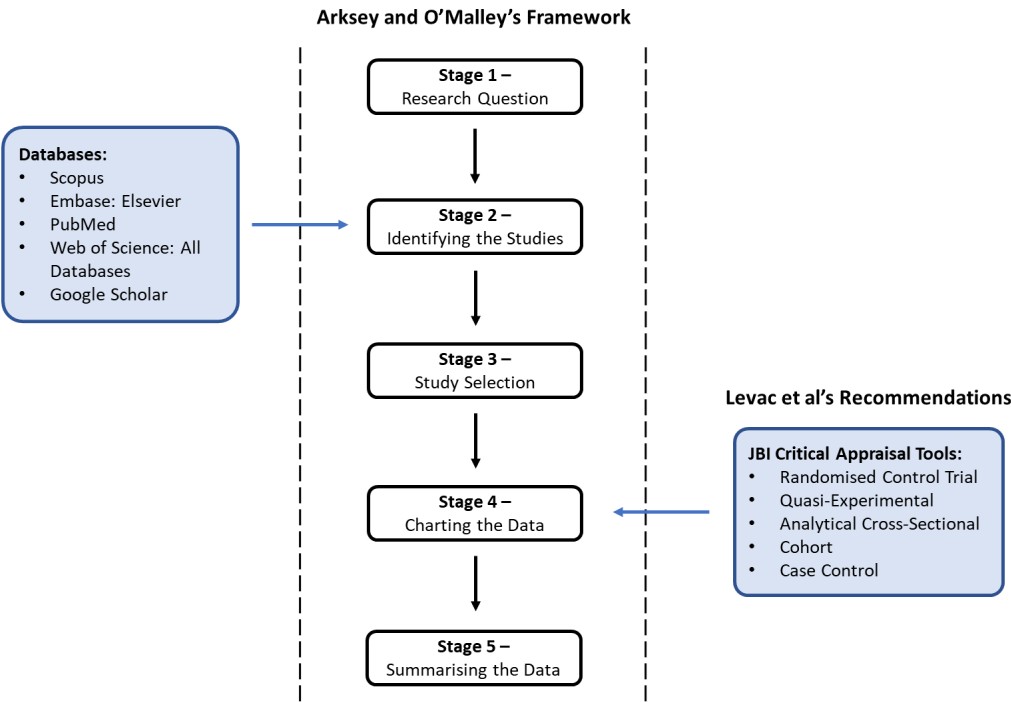

**Figure 2** Outline of the study methodology. JBI, Joanna Briggs Institute.

**Table 1** Review definitions

| Term | Definition |
|---|---|
| Post-traumatic osteoarthritis | Joint disorder characterised by extracellular matrix degradation initiated by micro-injury and macro-injury resulting in the manifestation of molecular derangement (eg, abnormal joint tissue metabolism) followed by anatomical, and/or physiological derangements (eg, cartilage degradation), that can culminate in illness.[24] In this review, we will focus on the disease component of the disorder. |
| Intra-articular knee injury | Previous research has identified cruciate ligament and meniscus injuries have the highest risk of developing into post-traumatic osteoarthritis.[20] Considering these are also the most common types of intra-articular knee injury, our review will focus on both ligament and meniscus injuries only. |
| Physical activity | Any bodily movement produced by skeletal muscles that results in energy expenditure.[146] Further, we will include the term 'exercise' within this definition of physical activity, as although they describe different concepts, they are often used as synonyms.[146] Exercise is defined as a subset of physical activity that is planned, structured, and repetitive and has as a final or an immediate objective of improvement or maintenance of physical fitness.[146] |
| Systemic inflammation | A consensus definition on the type, number and value of inflammatory biomarkers for systemic inflammation diagnosis has not been devised. In this review, we will use concentrations of key circulating proteins and cytokines identified in previous research as being involved with inflammation processes and osteoarthritis.[147–152] |
| Knee joint load | The research team defines knee joint load as quantification of the forces and/or moments applied to the knee joint specifically. |
| Strength (lower body) | Strength is the ability to produce a maximal voluntary muscular contraction against an external resistance.[153] We consider the following movements as relevant to this review:<br>▶ Hip extension/flexion/abduction/adduction.<br>▶ Knee extension/flexion/abduction/adduction.<br>▶ Ankle dorsi/plantar flexion, supination/pronation. |
| Adipose tissue | Specialised loose connective tissue that is extensively laden with adipocytes.[154] Adipose tissue is arranged in discrete depots,[155] and we consider the following depots at any body location relevent to this review:<br>▶ Subcutaneous.<br>▶ Intraperitoneal/viseral.<br>▶ Intermuscular (considered to be similar to intraperitoneal/visceral).[156] |
| Intramuscular adipose tissue (lower body) | Intramuscular adipose tissue is the visible fat found within a muscle.[156] We consider the intramuscular adipose content in any of the following muscle groups/muscles relevant to this review:<br>▶ Gluteal.<br>▶ Hamstring.<br> – Bicep femoris.<br> – Semitendinosus.<br> – Semimembranosus.<br>▶ Quadriceps.<br> – Vastus lateralis.<br> – Rectus femoris.<br> – Vastus medialis.<br> – Vastus intermedius.<br>▶ Tibialis anterior.<br>▶ Gastrocnemius.<br>▶ Soleus. |
| Muscle size (lower body) | Muscle size if the dimension of skeletal muscle, which can be devised into a multiscale[157] consisting of;<br>▶ Organ.<br>▶ Tissue.<br>▶ Cellular.<br>▶ Subcellular.<br>▶ Molecular.<br>We consider the muscle size of muscle groups/muscles relevant to this review:<br>▶ Gluteal.<br>▶ Hamstring.<br> – Bicep femoris.<br> – Semitendinosus.<br> – Semimembranosus.<br>▶ Quadriceps.<br> – Vastus lateralis.<br> – Rectus femoris.<br> – Vastus medialis.<br> – Vastus intermedius.<br>▶ Tibialis anterior.<br>▶ Gastrocnemius.<br>▶ Soleus. |
| Bone mineral content | The research team defines bone mineral content as the amount of minerals (mostly calcium and phosphorus) contained in bone.<br>Total bone mineral content of the body is considered relevant to this review. Additionally, the mineral content of the following bones is also considered relevant to this review:<br>▶ Femur.<br>▶ Tibia.<br>▶ Fibular. |

adiposity, strength (lower body), intra-muscular adipose tissue (lower body), muscle size (lower body), and bone mineral content. For this review, we will use consensus definitions for intra-articular knee injury, PA, the themes underpinned by PA, and PTOA (table 1). If there is no agreed or clear definition, then best practices used by previous literature or consensus among the research team will be adopted.

## Stage 2: identifying relevant studies

The search terms include population, independent variable, and outcome (online supplemental material 2). Using the Peer Review of Electronic Search Strategy (PRESS)[104] guidelines, a professional health science librarian was consulted to develop and tailor search strategies for each database. Electronic databases Scopus, Embase: Elsevier, PubMed, Web of Science: all databases, and Google Scholar will be searched. Literature published between 1970 and 2022 will be included, covering the period during which clinical knee injury diagnosis was formalised and surgical reparative techniques were popularised.[105 106] Primary research studies and grey literature will be included.[93 107 108]

## Stage 3: study selection

Inclusion and exclusion criteria are presented in table 2.

Outcome measures for PTOA disease must include medical imaging or molecular indicators of joint health (eg, metabolic, proteomic, genomic, transcriptomic).[24] Imaging diagnosis of disease may include radiography or magnetic resonance imaging (MRI).[109–113] Imaging indicators for disease processes include MRI[114–117] and ultrasound techniques.[118–120] For molecular indicators of disease processes, OsteoArthritis Research Society International (OARSI) recommended biomarkers[121] will be included alongside other biomarkers of cartilage metabolism which have been deemed of interest in previous reviews due to their association with joint health (online supplemental material 3).[122–125] This list of outcome measures is not exhaustive and will be added to if other relevant outcomes are identified during Stage 2: identifying relevant studies.

To establish the face and content validity of the inclusion and exclusion criteria, and the selected outcome measures, feedback was sought from a surgeon, an osteopath, a physiotherapist, a physiologist and a biomechanist independent from the research team, with experience ranging from clinical practice to sport research. These inclusion and exclusion criteria will be applied to title/abstract screening, which will be conducted using Covidence (Covidence Systematic Review Software, Veritas Health Innovation, Melbourne, Australia. Available at www.covidence.org). Prior to screening start, JCo, JCa and MF will be familiarised and trained[126] by the lead author (KM) to conduct title/abstract screening using a screening tool hosted on Research Electronic Data Capture platform (REDCap)[127] (online supplemental material 4). Following the training session, inter-rater reliability of the inclusion and exclusion criteria for title/abstract screening will be assessed based on a random sample of 120 titles/abstracts. Percentage agreement and Cohen's kappa (κ)[128] will be determined between all reviewers (KM—100% of records, JCo—33%, JCa—33%, MF—33%). The inclusion and exclusion criteria will be deemed reliable if the agreement is 'strong' (percentage agreement >63% and κ >0.60),[129] and title/abstract screening will only begin once the agreement level is

**Table 2** Review inclusion and exclusion criteria

| Inclusion | Exclusion |
|---|---|
| Includes participants with a history of experiencing intra-articular injury or reparative surgery for the injury. | Reporting of any other medical condition (disease, illness, or disorder): <br>► Musculoskeletal. <br>► Immunological. <br>► Cardiological. <br>► Respiratory. <br>► Neurological. <br>► Metabolic. |
| Measures any association between PA or factors determined by PA (systemic markers of inflammation, joint loading, adiposity, muscle strength, lower body muscle size, intermuscular fat, bone density), and either post-traumatic osteoarthritis diagnosis or indicators of joint degeneration. | Evidence of intra-articular injury incidence to any of the following joints prior to or following intra-articular knee injury: <br>► Ankle. <br>► Knee. <br>► Hip. <br>► Shoulder. <br>► Elbow. |
| 18–40 years of age | Revision surgery if undergone initial reparative surgery. |
| Primary research studies, grey literature | Animal models, in vitro studies, purely in silico studies with no human component (ie, simulated data with no data capture element working with humans), meta-analyses, narrative reviews, systematic reviews, scoping reviews, protocols, commentaries, position or consensus statements, and purely cadaveric studies. |
| Human research | |
| Published in English | |
| Literature published between 1970 and 2022 | |
| PA, physical activity. | |

strong. Following training and provided 'strong' agreement between reviewers is achieved, title/abstract screening will begin. During this process reviewing pairs (KM—100% of records, JCo—33%, JCa—33%, MF—33%) will independently screen all titles and abstracts to reduce the quantity of errors.[130] If disagreements cannot be resolved within reviewing pairs, a consensus decision across the wider research team will be made following a discussion. Duplicate results returned during title/abstract screening will be removed using Covidence.

Following title and abstract screening, full-text screening will be conducted by reviewing pairs (KM—100% of records, JCo—33%, JCa—33%, MF—33%).[126] Prior to full-text screening, JCo, JCa and MF will be familiarised

and trained[126] by the lead author (KM) to conduct full text screening using a screening tool hosted on REDCap[127] (online supplemental material 5). Following the training session, inter-rater reliability of the inclusion and exclusion criteria for the full-text screening will be assessed based on a random sample of 20 full texts. Once 'strong' agreement between reviewers is achieved, full text screening will begin. All full texts will be independently screened (KM—100% of records, JCo—33%, JCa—33%, MF—33%) to reduce quantity of errors.[130] If disagreements cannot be resolved within reviewing pairs, a consensus decision across the research team will be made following a discussion. In line with common practice[131 132] and best-practice recommendations,[133] we will contact corresponding and coauthors via email if details of a study are unclear during title/abstract or full-text screening (online supplemental material 6). A maximum of two attempted contacts will be made. The number of corresponding authors contacted, coauthors contacted, replies, and whether adequate information was provided, will be reported. Following full text screening, the remaining relevant studies to the review will undergo forward/backward screening.

## Quality and critical appraisal

In accordance with best-practice recommendations,[99] we will assess methodological quality before conducting data charting of included studies. As this review will include various study designs, we will employ Joanna Briggs Institute (JBI) appraisal tools respective to each study design (online supplemental material 7).[134–137] Individual items within each JBI appraisal tool will be assigned either a 'yes' (1), 'no' (0), or 'unclear' (seek clarification/further information) response to questions relating to the research quality. If study details are unclear, corresponding authors will be contacted following the same process as described in Stage 3: study selection (online supplemental material 5). A quality percentage score will be calculated, with smaller scores indicating lower quality and larger scores indicating higher quality. Identified studies will be included regardless of the quality appraisal score. The appraisals will be conducted by reviewing pairs (KM—100% of records, JCo—33%, JCa—33%, MF—33%) using REDCap.[127]

## Stage 4: charting the data

Data charting will be conducted in REDCap[127] by reviewing pairs (KM—100% of records, JCo—33%, JCa—33%, MF—33%) to reduce the risk of errors in data extraction.[130] Data charting will include author(s), country of study, study year, study design, participant age and sex/gender distributions, injury type, surgery type, type of injury management, time since injury/surgery, sporting history before injury (type and level), aim of study, independent variables, supervision level (if intervention), quantity (if intervention), adherence rate (if intervention), drop out (if intervention), outcome variables, who assessed the measures, statistical methodology, and the strength of the treatment response or relationship between variables. This form (online supplemental material 8) will be pilot tested by two reviewers and necessary amendments will be made if required (eg, other relevant measurement tools or outcome variables). If study details are unclear, corresponding authors will be contacted following the same process as described in Stage 3: study selection.

## Stage 5: summarising the data

Data will be summarised in three steps.[99] The first step will present the results as a descriptive numerical summary which will include: number of studies on each theme of the research question; frequency of study design; definitions and variables used for joint disease; overview of methodological quality of studies. The second step will present the results as individual sections for each theme of the research question. Each theme will be structured as follows:

► Strength of the treatment response or relationship between variables[138–141] (online supplemental material 9) for studies presented descriptively.
► In accordance with a previous scoping review on anterior cruciate ligament patients,[142] we will use the GRADE[89–91] (incorporates Bradford Hill criteria)[143–145] for quality of evidence for either a protective or degenerative, or if there is an unknown effect (online supplemental material 10). Study-by-study results will be presented in supplementary material, while the overall summary of results for each theme will be displayed within the article in a summary of findings table.

The third step will be to discuss and consider the meaning of the findings as they relate to the overall review purpose. We will also discuss implications for future PTOA research.

## ETHICS AND DISSEMINATION

Ethical approval is not required as this review maps and synthesises data generated from published literature. This review will be submitted for publication in a peer-reviewed sports medicine journal, irrespective of findings which indicate a positive or negative association between PA and joint health. Knowledge synthesised by this review will be presented at scientific conferences and disseminated via social media platforms.

## CONCLUSION

This study seeks to address a salient topic for people who experience a traumatic intra-articular knee injury and clinicians working with this population. Using a scoping review methodology to examine the known influence of PA on PTOA pathogenesis is novel and appropriate, considering the complexity of the topic and the multidisciplinary evidence base. This research may inform future PTOA research by helping to guide research

priorities and identifying outcome measures for clinical interventions.

**Acknowledgements** We thank Peter Bradley, the University of Bath's Subject Librarian for Health and Social and Policy Sciences for his help in developing the search terms and his role when implementing the Peer Review of Electronic Search Strategy (PRESS)116 guidelines. We also thank Dr David Bruce, Dr Nicos Haralabidis, Rosy Hyman, Charli Robertson and Lee Page for their expert feedback developing the inclusion/exclusion criteria and outcome measures.

**Contributors** KM: Conceptualisation, Methodology, Writing—Original Draft, Writing—Review and Editing. JCo: Methodology, Writing—Review and Editing. MF: Methodology, Writing—Review and Editing. JCa: Methodology, Writing—Review and Editing. DC: Writing—Review and Editing, Supervision. J-PW: Writing—Review and Editing, Supervision. CM: Conceptualisation, Methodology, Writing—Review and Editing, Supervision.

**Funding** This work was funded by the University of Bath (funding refence: EA-FH1112) in affiliation with the Centre for Sport, Exercise and Osteoarthritis Research Versus Arthritis, University of Bath (grant reference: 21595).

**Competing interests** None declared.

**Patient and public involvement** Patients and/or the public were not involved in the design, or conduct, or reporting, or dissemination plans of this research.

**Patient consent for publication** Not applicable.

**Provenance and peer review** Not commissioned; externally peer reviewed.

**ORCID iD**
Karl Morgan http://orcid.org/0000-0002-4642-9318

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
