## [Reviewer comments · BMJ Open]

ARTICLE DETAILS

TITLE (PROVISIONAL)	Understanding the Role of Physical Activity on the Pathway from Intra-Articular Knee Injury to Post-Traumatic Osteoarthritis Disease in Young People: A Scoping Review Protocol
AUTHORS	Morgan, Karl; Cowburn, James; Farrow, Matthew; Carter, Josh; Cazzola, Dario; Walhin, Jean-Philippe; McKay, Carly

VERSION 1 – REVIEW

REVIEWER	D'Ambrosi, Riccardo IRCCS Istituto Ortopedico Galeazzi
REVIEW RETURNED	24-Aug-2022

GENERAL COMMENTS	title: be more clear about the aim of the study Introduction: be more specific what is the aim and the hypothesis? how can you get this? methods who will perform the research? intro really too long focus on the controversies of the topic and the ratio what is new? what add to current literature? what is your hypothesis? methods who will perform the research? what are the primary outcomes? it's not clear how the results will be presented, and what do you expect conclusions overall coherent
--

REVIEWER	Mündermann, Annegret University Hospital Basel, Orthopaedics and Traumatology
REVIEW RETURNED	24-Oct-2022

GENERAL COMMENTS	Title: Understanding the Role of Physical Activity on the Pathway from Intra-Articular Knee Injury to Post-Traumatic Osteoarthritis in Young People: A Scoping Review Protocol General Comments
---

	The goal of this paper is to describe the protocol for a scoping review on identifying and presenting available empirical evidence regarding the association between physical activity (PA) and joint degeneration after intra-articular knee injury, summarising the evidence using an adapted Grading of Recommendations, Assessment, Development and Evaluations, identifying potential mechanistic pathways through which PA could influence posttraumatic osteoarthritis (PTOA) pathogenesis, and highlighting gaps in current understanding of the association between PA and joint degeneration following joint injury. The manuscript is very well written, and the design of the protocol thorough. Nonetheless, the authors should address some minor points: Minor comments  1. The authors should present a timeline of the review and / or clearly state the current stage of the review. Has the screening process been started / completed? 2. The authors cite a narrative review (reference 61). Please clearly state the novel contribution of the proposed review. 3. Physical activity is not abbreviated in all instances in the manuscript. Please ensure that abbreviations are used once introduced. 4. Line 86: and will adhere -> and adhere 5. Line 93: Please define what “young” means. It is specified in the table but I suggest to state it in the text. 6. It is unclear why the authors include bone mineral density as a topic of interest. 7. In the search strategy, the authors included MRI parameters of articular cartilage morphology and quality. Please also mention these somewhere in the manuscript. 8. I assume that the search result will be deduplicated. Please included this step in the flow diagrams. Also, will there be citation tracking? If yes, please state so; if no, then please state a reason why this is not done. 9. In the search strategy, the authors include biomarker*. I suggest to also include “blood marker” or just *marker*. 10. What is the rationale for not including MMP-9? What about inflammatory markers such as TNF-alpha or IL-6? 11. The authors should consider including the EQ-5D as measurement tool for quality of life.
--	---

VERSION 1 – AUTHOR RESPONSE

Reviewer 1 (R1).Comment (C[Number])

R1.C1 title: be more clear about the aim of the study	Although we acknowledge the title is not as direct as it would be in perhaps an experimental study, we believe the title is appropriate. For example, the title reflects the exploratory nature of the scoping review whilst also incorporating best practice by including the type of review in the title. Ultimately, although we appreciate the feedback, we feel strongly the title is appropriate and a substantial change in title is not required. However, we have amended the title to reflect the disease component alone.
--	--

R1.C2 Introduction: be more specific	As osteoarthritis is a complex joint disorder, with a multi-faceted aetiology, and numerous existing definitions, we feel the need to thoroughly characterise our understanding and approach of the disorder. Furthermore, the relationship between physical activity and PTOA is extremely complex, with a number of mediating variables which need to be discussed to fully characterise how physical activity influences joint health after a knee injury.
R1.C3 what is the aim and the hypothesis?	The primary, secondary and tertiary aims of the research are reported in the abstract introduction (lines 5-11) and Revised Main Manuscript introduction (lines 53-59).
R1.C4 how can you get this?	Unfortunately, we are unsure at to what this refers to, so we are unable to address this comment or make any changes at present. We would appreciate it if the reviewer could clarify this point.
R1.C5 who will perform the research?	Due to word count restrictions, we think it prudent to not include the names of authors conducting the research in the abstract. Please see the initialised names of authors found in the Revised Main Manuscript, which is standard practice.
R1.C6 really too long	Whilst we understand the reviewer's concern, we feel that given the complexity of the condition, and the multi-faceted relationship between physical activity and osteoarthritis, greater characterisation of the topic is required. Furthermore, as it is a protocol paper, we feel this is an excellent opportunity to characterise why this review is required. Although we acknowledge the introduction is long, from the feedback provided by the reviewer is very difficult to understand what part of it should be shortened. Also, in relation to other sections of this protocol, we feel 782 words of a 2,670-word document (Revised Main Manuscript) is not excessive.
R1.C7 focus on the controversies of the topic and the ratio what is new?	Unfortunately, we are unsure at to what this refers to, so we are unable to address this comment or make any changes. We would appreciate it if the reviewer could clarify this point.
R1.C8 what add to current literature?	This review will significantly contribute to current understanding by addressing a salient topic for people who experience a traumatic intra-articular knee injury and clinicians working with this population. Using a scoping review methodology to examine the known influence of PA on PTOA pathogenesis is novel and appropriate, considering the complexity of the topic and the multi-disciplinary evidence base. This research may inform future PTOA research by helping to guide research priorities and identifying outcome measures for clinical interventions.
R1.C9 what is your hypothesis?	Given the nature of a scoping review, it would be inappropriate to incorporate a specific hypothesis. The purpose of a scoping review can be best defined as to: 'Aim to map the existing literature in a field of interest in terms of the volume, nature, and characteristics of the primary research (Arksey and O'Malley, 2005; Int. J. Soc. Res. Methodol.).' However, in line with best and common practice when conducting a scoping review, we have explicitly stated that this scoping review is guided by the question 'What is the role of PA on the trajectory from intra-articular knee injury to post-traumatic osteoarthritis in young men and women?'

R1.C10 who will perform the research?	Please see the initialised names of authors found in the Revised Main Manuscript.
R1.C11 what are the primary outcomes?	Please see lines in the Revised Main Manuscript 126-132 for a general description of the outcome measurements. We have also included a detailed list of outcome measures in Supplementary Material 3.
R1.C12 it's not clear how the results will be presented, and what do you expect	Please see lines 204-219. In accordance with best practice for scoping reviews, we outline the steps we will take when presenting our data. Furthermore, we will use an adapted Grading of Recommendations, Assessment, Development and Evaluations (GRADE). Inferring or speculating on the results of the GRADE panel would not be appropriate for this review.

Reviewer 2 (R2)

R2.C1 The authors should present a timeline of the review and / or clearly state the current stage of the review. Has the screening process been started / completed?	Please see lines 72-74 of the Revised Main Manuscript with the timeline now included.
R2.C2 The authors cite a narrative review (reference 61). Please clearly state the novel contribution of the proposed review.	We acknowledge the need for clarification of why this review, conducted with a systematic approach, is required. Please see lines 81-83 of the Revised Main Manuscript.
R2.C3 Physical activity is not abbreviated in all instances in the manuscript. Please ensure that abbreviations are used once introduced.	We thank the reviewer for highlighting this and apologise that the term 'physical activity' was not abbreviated throughout the original Main Manuscript. The abbreviations can now be found in the Revised Main Manuscript.
R2.C4 Line 86: and will adhere -> and adhere	We have made the recommended change.
R2.C5 Line 93: Please define what "young" means. It is specified in the table but I suggest to state it in the text.	Thank you for highlighting this issue, please see our definition of 'young' now included in the Revised Main Manuscript, (lines 102-103).
R2.C6 It is unclear why the authors include bone mineral density as a topic of interest.	We have included the bone mineral density considering the strong, positive association between bone mineral density and risk of osteoarthritis in idiopathic osteoarthritis (please see this review [https://www.ncbi.nlm.nih.gov/pmc/articles/PMC4303262/] and a prospective cohort [https://arthritis-research.biomedcentral.com/articles/10.1186/s13075-017-1314-0]). Furthermore, there is a strong association between physical activity and bone mineral density. Considering bone mineral density appears to be associated with both physical activity and the disease, we think it prudent to explore what is the current evidence linking these variables together in a PTOA context.
R2.C7 In the search strategy, the authors included MRI parameters of articular cartilage morphology and quality. Please also mention these somewhere in the manuscript.	Please refer to the Revised Main Manuscript lines 128-129. We feel that the sentence ' Imaging indicators for disease processes include MRI and ultrasound (US) techniques. ', demonstrates that medical imaging techniques, inclusive of parameters of articular cartilage morphology and quality will be included in this review.

R2.C8 I assume that the search result will be deduplicated. Please included this step in the flow diagrams. Also, will there be citation tracking? If yes, please state so; if no, then please state a reason why this is not done.	We thank the reviewer for their expert insight, and we have now amended the section to explicitly state results will be de-duplicated (please see lines 157-158). Additionally, we have now reported we will be conducting forward and backward searches (please see lines 173-174).
R2.C9 In the search strategy, the authors include biomarker*. I suggest to also include "blood marker" or just *marker*.	We agree with the reviewer that the term "blood marker" alone should be included and we thank them for their expertise. Please see the amended search terms where we have also included "urine marker" and "synovial marker".
R2.C10 What is the rationale for not including MMP-9? What about inflammatory markers such as TNF-alpha or IL-6?	We agree there was an oversight in our list in appropriate markers categorised as substrates/metabolites/gene expression of cartilage and bone. We have now amended our screening tools to reflect this. Regarding the inflammatory markers, such as TNF-alpha or IL-6 we have decided to separate these as causal mechanisms, rather than consequential responses. For example, in vitro , TNF-alpha can activate matrix metalloproteases which then cleaves type II collagen. As systemic inflammation is linked to both physical activity and adiposity, we feel this is a potential causal pathway, rather than a response specific to joint deterioration.
R2.C11 The authors should consider including the EQ-5D as measurement tool for quality of life.	We kindly ask the reviewer to please note that we have now adjusted our focus to entirely on the disease component of the joint disorder only. However, we would like to thank the reviewer for an excellent suggestion, and we will take their advice forward if we target the illness component in any further research/review conducted.

VERSION 2 – REVIEW

REVIEWER	D'Ambrosi, Riccardo IRCCS Istituto Ortopedico Galeazzi
REVIEW RETURNED	12-Jan-2023
GENERAL COMMENTS	after revisions the article deserve to be accepted

VERSION 2 – AUTHOR RESPONSE

Reviewer 1 (R1).Comment (C[Number])

ED.C1	Removal of Justification of Research from PPI Section (pages 6-7, lines 77-86 of marked version)
-------	--